# Assessing the reliability of abdominal adiposity measurements by transabdominal ultrasound in first trimester pregnancies with sonographers of varying experience levels

**Noppasin Khwankaew, Nattraporn Srisovanna, Ninlapa Pruksanusak** [ID]*

Department of Obstetrics and Gynecology, Faculty of Medicine, Prince of Songkla University, Hat Yai, Thailand

* nin056@hotmail.com

## Abstract

### Objective

Given the growing evidence linking increased abdominal adiposity in early pregnancy to a higher risk of developing gestational diabetes mellitus (GDM), this study aims to evaluate the reliability of the sonographic measurements of abdominal adiposity during the first trimester of pregnancy, comparing the performance of sonographers with varying levels of experience.

### Methods

A cross-sectional study was conducted on singleton Asian pregnant women with gestational ages between 11 and 14 weeks. Transabdominal ultrasound measurements of abdominal adiposity were performed by three sonographers with different experience levels, including measurements of subcutaneous fat thickness (SFT), visceral fat thickness (VFT), and preperitoneal fat thickness (PFT) using three techniques. Intra- and interobserver reliability were assessed using intraclass correlation coefficient (ICC) with a 95% confidence interval. Spearman pairwise correlation coefficients were used to analyze the correlation between maternal body composition and the sonographic measurements of abdominal adiposity.

### Result

Ninety pregnant women participated. The overall ICCs indicated excellent intraobserver reliability for all parameters, with VFT measurement showing slightly lower reliability for less experienced operators. Interobserver reliability was excellent for all measurements except VFT, which showed good reliability between the most and least experienced operators. VFT correlated significantly with maternal waist-to-hip ratio (WHR).

**Data availability statement:** All relevant data are within the manuscript and its Supporting Information files.

**Funding:** Prince of Songkla University.

**Competing interests:** The authors have declared that no competing interests exist.

## Conclusion

Sonographic measurements of abdominal adiposity during the first trimester show high reliability, particularly when performed by experienced sonographers. Notably, VFT measurement displayed a significant positive correlation with maternal WHR, highlighting its potential as an indicator of central adiposity.

## Introduction

Obesity is a major health concern in many affluent societies today, including an increased risk for type 2 diabetes, dyslipidemia, heart disease, and hypertension. Importantly, this condition predisposes individuals to numerous pregnancy-related complications and long-term morbidity and mortality [1,2]. The body mass index (BMI) is most often used to diagnose obesity, with different criteria for Asian and non-Asian populations [3]. Using these classifications, the trend of obesity increases with age and varies among ethnicities [4,5], paralleling the rising rates of gestational diabetes mellitus (GDM) in pregnant patients.

The prevalence of GDM is estimated to be approximately 14.7%, varying across different regions and ethnic groups [6]. Notably, certain ethnic groups, such as Asians, have been shown to have a higher risk of developing GDM compared to others [7]. In Thailand, for instance, the prevalence of GDM surged from 3.4% in 2003 to 22.0% in 2022 [5]. Interestingly, more than half (57.3%) of GDM patients in our country had a BMI of less than 25 kg/m². Similarly, at our institute (Songklanagarind Hospital, a quaternary care center in Southern Thailand), pregnant women with GDM had an average BMI of 24.4. These findings highlight the need for tailored screening and management strategies. In particular, incorporating additional measures such as sonographic assessment of abdominal adiposity may serve as a valuable complement to existing GDM screening tools.

It is well known that central adiposity, characterized by excess fat accumulation around the trunk, increases the risk of cardiovascular disease, hypertension, and diabetes more than peripheral adiposity. This is particularly relevant in the context of GDM, especially in non-obese patients categorized by BMI range, as central adiposity is associated with insulin resistance and an increased risk of adverse outcomes [8]. While BMI is commonly used as a screening tool for obesity, it provides limited information on the distribution of body fat, which is known to influence metabolic health [9]. Computerized tomography (CT), Magnetic resonance imaging (MRI), and body densitometry are considered better markers for central obesity than BMI, but they are impractical as screening tools during pregnancy [10]. However, accurately assessing central adiposity during pregnancy can be challenging by ultrasound. Although it is safe for the fetus, there is limited evidence regarding its utility as a predictor of GDM, especially in Asian populations, and concerning measurement sites and techniques.

Previous studies have suggested that first-trimester ultrasound measurements of abdominal adiposity, including subcutaneous fat thickness (SFT), visceral fat

thickness (VFT), and preperitoneal fat thickness (PFT) with different techniques, may offer valuable insights into the metabolic status of pregnant women and support early identification of those at risk of developing GDM [11–13]. However, there remains a need for further research to establish the reliability and validity of these ultrasound measurements [13], particularly in diverse populations such as Asian women.

Therefore, this study aims to evaluate the reliability of sonographic measurements of abdominal adiposity in Asian pregnant women during the first trimester using three different measurement techniques. By addressing this knowledge gap, we aim to encourage further research and contribute to the development of an adiposity-based risk assessment tool as a complementary approach to existing GDM screening strategies. Such a tool could be tailored to the specific characteristics of Asian populations and ultimately help improve maternal and fetal outcomes.

## Materials and methods

This cross-sectional study was conducted at the Maternal-Fetal-Medicine (MFM) unit within the Obstetric and Gynecologic department of Songklanagarind Hospital between 6/10/2023 and 6/6/2024. The study was approved by the Institutional Review Board of the Faculty of Medicine, Prince of Songkla University (REC.66-369-12-1). Singleton Asian pregnant women with a gestational age between 11 and 14 weeks who were attending routine first-trimester screening at the MFM unit were invited to participate. They provided written inform consent after receiving detailed information about the study objectives and procedures. Exclusion criteria included pre-existing metabolic diseases such as diabetes mellitus, hypertension, dyslipidemia, and cardiovascular disease.

Three sonographers with different levels of experience conducted the ultrasonography: operator 1, NP, a senior MFM staff member with over 10 years of experience in ultrasound; operator 2, NK, a young MFM staff member with less than 5 years of experience; and operator 3, NS, a first year MFM fellow. Transabdominal 2D ultrasonography was performed using a GE Voluson E10 equipment (GE Medical Systems, Zipf, Austria) and a Samsung V8 (Samsung Medison Co., Ltd, Seoul, Republic of Korea) with a 2.0–5.0 MHz curvilinear probe. Each participant was independently scanned by two operators in the following pairs (operator 1 and 2, operator 1 and 3, or operator 2 and 3). Based on prior studies [12,14,15], measurements focused on SFT using three techniques in the midline of the abdomen including technique 1 involved a transverse scan at the level of linea alba about 1–2.5 centimeters above umbilicus, technique 2 used a longitudinal scan at the cervix-placenta view, with key anatomical landmarks including the bladder, cervix, and uterus clearly visible, and technique 3 employed a longitudinal scan at the level of the linea alba viewing the anterior surface of the liver. Measurements also included VFT with technique 1, PFT with technique 3. The specific measurements for each technique were as follows:

Technique 1 (Fig 1):

- SFT1: Vertical distance from the skin surface to the outer border of the linea alba.

- VFT: Vertical distance from the internal layer of the rectoabdominal muscle and the anterior wall of the aorta and perpendicular to the aorta.

Technique 2 (Fig 2):

- SFT2: Vertical distance from the skin surface to the peritoneal fascia. Three measurements were obtained including SFT 2.1, 2.2, and 2.3 that measures at the midline of the ultrasound probe, 5 mm to the left of the midline, and 5 mm to the right of the midline, respectively.

Technique 3 (Fig 3):

- SFT3: Vertical distance from the skin surface to the anterior edge of the preperitoneal fat.

- PFT: Vertical distance from the anterior edge of the preperitoneal fat to the anterior surface of the liver.

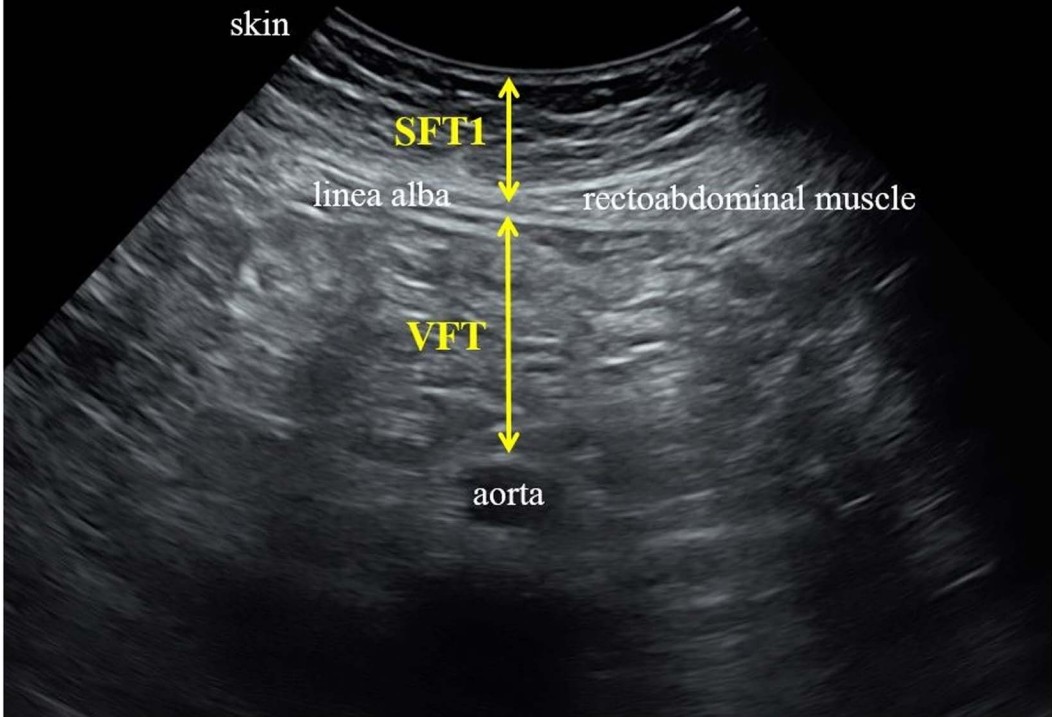

SFT1: Subcutaneous fat thickness 1, VFT: Visceral fat thickness

**Fig 1. Ultrasound image showing measurement of abdominal adiposity using Technique 1.**

Ultrasound measurements were taken with care to avoid excessive pressure that might artificially compress the abdominal tissue. The first operator performed the measurements with three independent measurements of each parameter, then the second operator conducted the scan independently on the same subject on the same day. Each operator was blinded to the results of the other. Average values were calculated to analyze interobserver reliability..

Data on intra- and interobserver reliabilities were reported using the intraclass correlation coefficient (ICC) with a 95% confidence interval. The interpretation of the ICC was as follows: 0.00–0.20, poor reliability; 0.21–0.40, fair reliability; 0.41–0.60, moderate reliability; 0.61–0.80, good reliability; and 0.81–1.00, excellent reliability. To assess reliability, Bland-Altman plots were first used, followed by modifications to establish the limits of agreement (LOA), which represent the range of variability in differences and indicate test-retest reliability for 95% of the population. Additionally, Spearman pairwise correlation coefficients were used to analyze the correlation between maternal body fat and sonographic measurements of abdominal adiposity. Statistical significance was set at a p-value of <0.05, and analyses were performed using Stata, version 14.2 (StataCorp, College Station, TX, USA).

## Results

Ninety singleton pregnant women were enrolled with a median age of 31 years. The maternal body fat measurements, including BMI, waist circumference (WC), hip circumference (HC), and their ratio, waist-to-hip circumference ratio (WHR), are shown in Table 1. The median BMI and WHR were 21.5 kg/m² and 0.87, respectively. Regarding the sonographic measurement of abdominal adiposity, the thickest adipose tissue measured was the VFT. SFT using technique 2 at the cervix-placenta view had a similar value in three areas and was thicker compared to techniques 1 and 3.

   

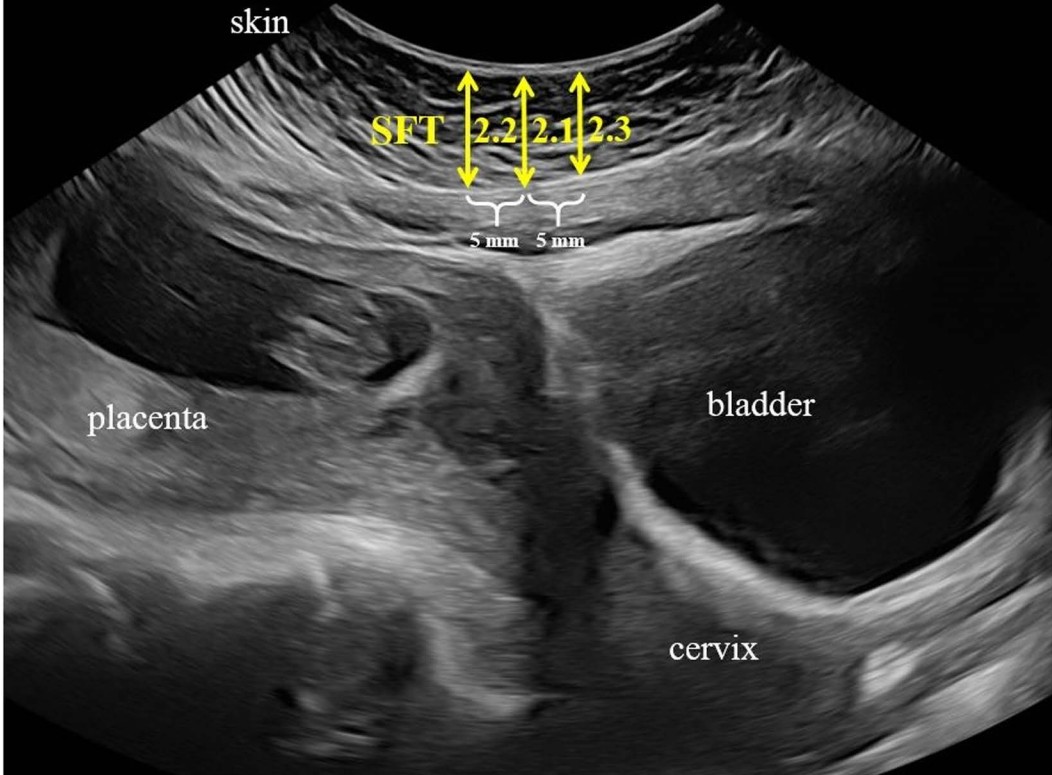

SFT2.1: Subcutaneous fat thickness 2.1, SFT2.2: Subcutaneous fat thickness 2.2,
SFT2.3: Subcutaneous fat thickness 2.3

**Fig 2. Ultrasound image showing measurement of abdominal adiposity using Technique 2.**

Intraobserver reliabilities of three operators are presented in Table 2. Excellent reliabilities were found in all sonographic measurements of abdominal adiposity, although the less experienced operator has lower reliability for VFT measurements. Additionally, interobserver reliabilities for each operator with different level of experience are shown in Table 3. The ICC indicated excellent reliabilities in all abdominal adiposity parameters, except for VFT measurements between the most and least experienced operators, which showed good reliability.

There was a significant positive correlation between all sonographic measurements and BMI, WC, and HC. For the WHR, only VFT showed a significant positive correlation, whereas other sonographic measurements of abdominal adiposity showed no significant correlation (Table 4, Fig 4).

## Discussion

Regarding the sonographic measurements of abdominal adiposity taken by three operators with different levels of experience, the intraobserver and interobserver reliabilities showed excellent consistency in all parameters, except for VFT measurements between the more and less experienced operators, which showed good reliability. Interestingly, only the VFT sonographic measurement had a significant positive correlation with the maternal WHR.

In comparing the agreement of SFT sonographic measurements within each operator and between two operators with different levels of experience, our study confirmed acceptable results for all three techniques. These techniques were

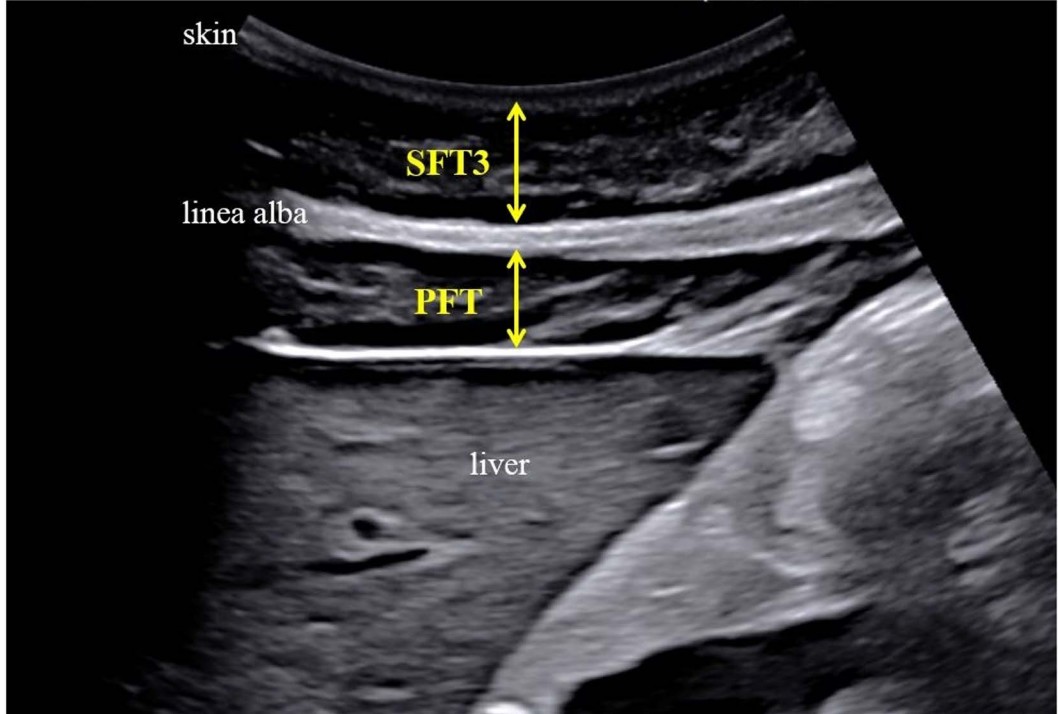

SFT3: Subcutaneous fat thickness 3, PFT: Preperitoneal fat thickness

**Fig 3. Ultrasound image showing measurement of abdominal adiposity using Technique 3.**

**Table 1. Demographic characteristics.**

| Parameters | Value |
|---|---|
| Maternal age (years), mean (SD) | 31 (4.63) |
| GA at USG (weeks), median (IQR) | 12 (12–13) |
| BMI at USG (kg/m²), median (IQR) | 21.5 (20–24) |
| Weight circumference (cm), median (IQR) | 80 (75–86) |
| Hip circumference (cm), median (IQR) | 92 (88–100) |
| Weight to hip ratio, median (IQR) | 0.87 (0.83–0.89) |
| Measurements of abdominal adiposity | |
| Subcutaneous fat thickness (SFT), median (IQR) | |
| Subcutaneous fat thickness 1 (SFT1) | 17.55 (14.00–20.60) |
| Subcutaneous fat thickness 2 (SFT2) | |
| Subcutaneous fat thickness 2.1 (SFT2.1) | 19.95 (18.00–23.68) |
| Subcutaneous fat thickness 2.2 (SFT2.2) | 20.35 (18.10–23.90) |
| Subcutaneous fat thickness 2.3 (SFT2.3) | 20.00 (18.00–23.70) |
| Subcutaneous fat thickness 3 (SFT3) | 14.10 (11.53–18.20) |
| Visceral fat thickness (VFT), median (IQR) | 33.15 (24.23–40.70) |
| Preperitoneal fat thickness (PFT), median (IQR) | 10.25 (8.00–12.90) |
| Total abdominal fat thickness (TAT), median (IQR) | 49.55 (40.40–59.60) |

BMI: body mass index, cm: centimeter, GA: gestational age, IQR: interquartile range, kg: kilograms, m: meter, SD: standard deviation, USG: ultrasound.

**Table 2. Intraobserver variabilities of the sonographic measurements of abdominal adiposity.**

| Parameters | Intraclass correlation coefficient (95% CI) | | |
|---|---|---|---|
| | Operator 1 | Operator 2 | Operator 3 |
| Subcutaneous fat thickness (SFT) | | | |
| Subcutaneous fat thickness 1 (SFT1) | 0.95 (0.93–0.97) | 0.98 (0.97–0.99) | 0.97 (0.95–0.98) |
| Subcutaneous fat thickness 2 (SFT2) | | | |
| Subcutaneous fat thickness 2.1 (SFT2.1) | 0.94 (0.92–0.96) | 0.97 (0.96–0.98) | 0.93 (0.89–0.95) |
| Subcutaneous fat thickness 2.2 (SFT2.2) | 0.92 (0.88–0.95) | 0.97 (0.95–0.98) | 0.92 (0.87–0.95) |
| Subcutaneous fat thickness 2.3 (SFT2.3) | 0.92 (0.89–0.95) | 0.97 (0.95–0.98) | 0.95 (0.93–0.97) |
| Subcutaneous fat thickness 3 (SFT3) | 0.95 (0.93–0.97) | 0.98 (0.98–0.99) | 0.97 (0.96–0.98) |
| Visceral fat thickness (VFT) | 0.95 (0.92–0.97) | 0.99 (0.98–0.99) | 0.87 (0.80–0.92) |
| Preperitoneal fat thickness (PFT) | 0.94 (0.91–0.96) | 0.97 (0.95–0.98) | 0.95 (0.92–0.97) |

CI, confidence interval.

**Table 3. Interobserver variabilities of the sonographic measurements of abdominal adiposity.**

| Parameters | Operator 1 vs 2 | | Operator 1 vs 3 | | Operator 2 vs 3 | |
|---|---|---|---|---|---|---|
| | ICC (95% CI) | LOA (mm) | ICC (95% CI) | LOA (mm) | ICC (95% CI) | LOA (mm) |
| Subcutaneous fat thickness (SFT) | | | | | | |
| Subcutaneous fat thickness 1 (SFT1) | 0.93 (0.85–0.96) | −3.80–4.28 | 0.91 (0.86–0.95) | −4.25–4.17 | 0.97 (0.95–0.98) | −2.46–2.38 |
| Subcutaneous fat thickness 2 (SFT2) | | | | | | |
| Subcutaneous fat thickness 2.1 (SFT2.1) | 0.91 (0.82–0.95) | −3.71–4.58 | 0.89 (0.82–0.93) | −4.37–5.00 | 0.98 (0.97–0.99) | −1.82–1.74 |
| Subcutaneous fat thickness 2.2 (SFT2.2) | 0.86 (0.60–0.94) | −3.03–6.11 | 0.83 (0.71–0.90) | −4.48–6.66 | 0.98 (0.97–0.99) | −2.03–1.78 |
| Subcutaneous fat thickness 2.3 (SFT2.3) | 0.90 (0.79–0.95) | −4.63–4.35 | 0.85 (0.76–0.91) | −5.44–5.67 | 0.98 (0.97–0.99) | −2.05–1.88 |
| Subcutaneous fat thickness 3 (SFT3) | 0.97 (0.94–0.99) | −2.69–3.09 | 0.94 (0.91–0.97) | −3.79–4.29 | 0.92 (0.86–0.95) | −4.24–3.86 |
| Visceral fat thickness (VFT) | 0.86 (0.71–0.93) | −11.99–12.60 | 0.73 (0.51–0.86) | −15.21–15.96 | 0.85 (0.72–0.93) | −13.32–10.80 |
| Preperitoneal fat thickness (PFT) | 0.96 (0.81–0.98) | −2.38–1.06 | 0.97 (0.93–0.98) | −2.19–2.37 | 0.96 (0.92–0.98) | −1.48–1.89 |

CI, confidence interval; ICC, intraclass correlation coefficient; LOA, limit of agreement; mm, millimeter.

**Table 4. Correlations coefficients between the sonographic measurements of abdominal adiposity and maternal body fat measurement.**

| Parameters | Body mass index (BMI) | Waist circumference (WC) | Hip circumference (HC) | Weight to hip ratio (WHR) |
|---|---|---|---|---|
| Subcutaneous fat thickness (SFT) | | | | |
| Subcutaneous fat thickness 1 (SFT1) | 0.57 (0.00) | 0.51 (0.00) | 0.52 (0.00) | 0.26 (0.91) |
| Subcutaneous fat thickness 2 (SFT2) | | | | |
| Subcutaneous fat thickness 2.1 (SFT2.1) | 0.64 (0.00) | 0.61 (0.00) | 0.62 (0.00) | 0.27 (0.84) |
| Subcutaneous fat thickness 2.2 (SFT2.2) | 0.64 (0.00) | 0.60 (0.00) | 0.61 (0.00) | 0.27 (0.84) |
| Subcutaneous fat thickness 2.3 (SFT2.3) | 0.64 (0.00) | 0.61 (0.00) | 0.61 (0.00) | 0.29 (0.70) |
| Subcutaneous fat thickness 3 (SFT3) | 0.63 (0.00) | 0.53 (0.00) | 0.65 (0.00) | 0.15 (1.00) |
| Visceral fat thickness (VFT) | 0.52 (0.00) | 0.53 (0.00) | 0.43 (0.00) | 0.38 (0.04) |
| Preperitoneal fat thickness (PFT) | 0.60 (0.00) | 0.44 (0.00) | 0.45 (0.00) | 0.27 (0.88) |
| Total abdominal fat thickness (TAT) | 0.61 (0.00) | 0.60 (0.00) | 0.53 (0.00) | 0.37 (0.07) |

Adjusted Spearman pairwise correlation coefficients (p-value).

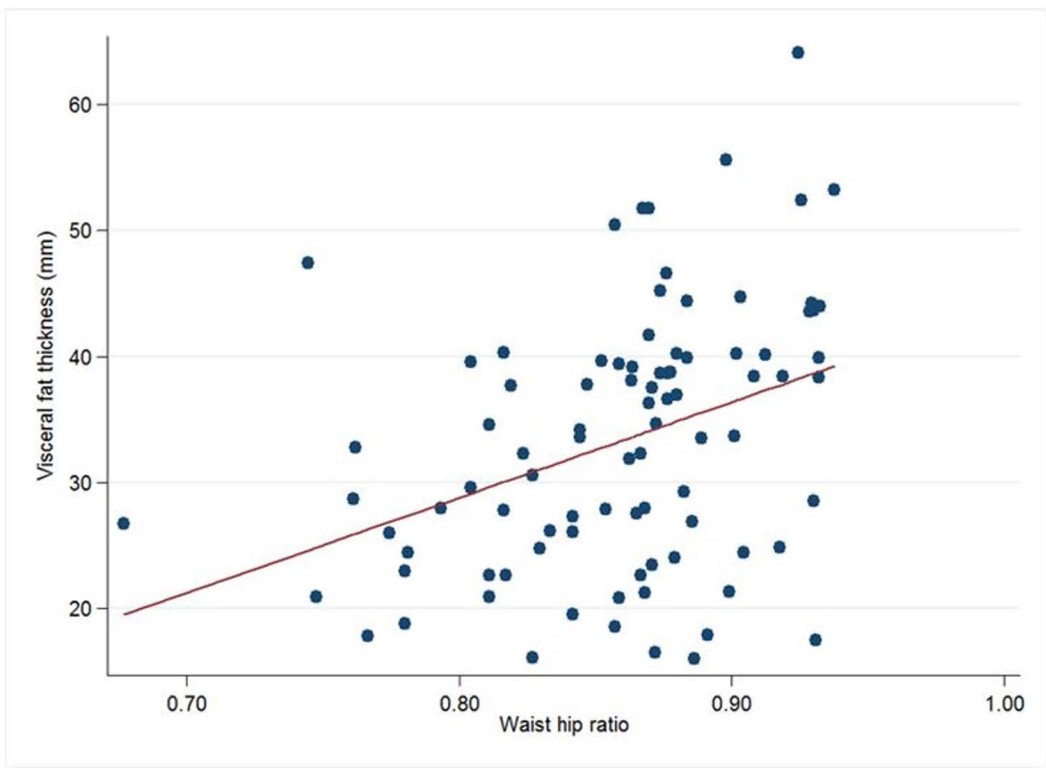

Spearman correlation coefficient = 0.38 (p-value = 0.04).

**Fig 4. Scatter plot of the sonographic measurements of visceral fat thickness and maternal weight to hip ratio.**

easily visible, less affected by the operators' experience, and consistent with the previous studies [11,12,16]. However, these previous studies did not identify the ultrasound experience of each operator and were conducted on Western and Australian populations. Our result was similar to a previous study which had a high intraobserver reliability (ICC 0.96), but it did not evaluate reliability between operators [17]. Based on our experience, all three SFT measurement techniques had minimal disturbance from maternal bowel movement, muscular movement during breathing, and pressure on adiposity by the ultrasound transducer, which explains why they demonstrated excellent reliability, even in less experienced hands. Although SFT had high reproducibility for measurement, its utility for predicting GDM showed contradictory results and did not appear significant in a recent meta-analysis [18]. In contrast to central adipose tissue, which requires meticulous measurement and has some limitations with less experienced operators, it may play an important role in the prediction of GDM.

Central adipose tissue measurements are proposed using two techniques: VFT and PFT. Our study demonstrated that PFT had excellent reliability both intra- and interobserver, while VFT had excellent reliability when performed by experienced operators. Focusing on VFT measurement with technique 1, we found excellent intraobserver reliability in all operators, similar to previous studies [19,20]. However, ours is the first study to evaluate interobserver reliability of this parameter, which showed high reliability only among moderate and highly experienced operators. Two main reasons that might obscure the view for measuring this parameter, especially for less experienced operator, were bowel gas, which is commonly found in pregnant women with bloating and/or nausea symptoms in the first trimester, and a full bladder during the measurement. Although the VFT had lower reliability than PFT for less experienced operators and was limited by

bowel gas and a full bladder, as discussed earlier, it is still a validated technique, as previous study has demonstrated a strong correlation between adipose tissue measurements by ultrasound and CT scans [21].

Our study found positive correlations between all sonographic abdominal adiposity measurements and maternal BMI, with correlation coefficients ranging from 0.52 to 0.64, which is comparable to previous studies. Kennedy et al. [16] found a moderate correlation (r = 0.56), in contrast to the higher agreement (r = 0.76) in the study by Nassr et al. [12]. However, BMI does not account for the relative contributions of central adipose tissue, while WHR correlates better with central adiposity, which is associated with an increased risk of maternal morbidity [22]. Interestingly, our findings reveal a significant positive correlation exclusively between maternal VFT and WHR, supporting VFT's representation of central adipose tissue. Furthermore, two recent meta-analyses [18,23] revealed that VFT could represent an additive factor in predicting the development of GDM when used in conjunction with other anthropometric or biological parameters or maternal risk factors. The maternal visceral fat tissue, encompassing the omentum as well as mesenteric fat, differs markedly from non-visceral fat areas. This adipose tissue drains directly to the liver via the portal venous system, thereby influencing hepatic inflow of free fatty acids and adipokines. This process precipitates the hepatic production of inflammatory mediators, which subsequently contributes to the metabolic risk observed in pregnant women [24].

The strength of this study is that it is the first to evaluate the reliabilities of abdominal adiposity sonographic measurements in different regions of pregnant women using three techniques, confirming the capacity to use these parameters with varying levels of operator experience. This ensures its applicability in real-world situations. However, for less experienced operators, we recommend additional training to implement these parameters effectively, especially in VFT measurement, which is significantly correlated with central adiposity deposition and might play an important role in predicting GDM in the future. Furthermore, we suggest that future and ongoing research should focus on the nomogram of adipose tissue in these regions to develop an effective model for predicting GDM in the next step.

## Conclusion

Sonographic measurements of abdominal adiposity during the first trimester show high reliability, particularly when performed by experienced sonographers. Notably, VFT measurement displayed a significant positive correlation with maternal WHR, highlighting their potential as an indicator of central adiposity.

## Supporting information

**S1 Data. Minimal data set.**
(XLSX)

## Acknowledgments

The authors wish to thank Alan Geater, PhD, of the Epidemiology Unit, Faculty of Medicine, Prince of Songkla University, for his valuable advice and assistance with data analysis.

## Author contributions

**Conceptualization:** Noppasin Khwankaew, Nattraporn Srisovanna, ninlapa pruksanusak.

**Data curation:** Noppasin Khwankaew, Nattraporn Srisovanna, ninlapa pruksanusak.

**Formal analysis:** Noppasin Khwankaew, ninlapa pruksanusak.

**Investigation:** ninlapa pruksanusak.

**Methodology:** Noppasin Khwankaew, Nattraporn Srisovanna, ninlapa pruksanusak.

**Project administration:** ninlapa pruksanusak.

**Writing – original draft:** Noppasin Khwankaew, ninlapa pruksanusak.

**Writing – review & editing:** ninlapa pruksanusak.

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
