## [Decision Letter · Decision Letter 0]

Dear Dr. pruksanusak,

Thank you for submitting your manuscript to PLOS ONE. After careful consideration, we feel that it has merit but does not fully meet PLOS ONE’s publication criteria as it currently stands. Therefore, we invite you to submit a revised version of the manuscript that addresses the points raised during the review process.

We look forward to receiving your revised manuscript.

Kind regards,

Athanasios G. Pantelis

Academic Editor

PLOS ONE

Journal Requirements:

“Prince of Songkla University”

5. We note that your Data Availability Statement is currently as follows: All relevant data are within the manuscript and in Supporting Information files.

Additional Editor Comments:

This is a scientifically sound and generally well-written manuscript on a novel subject. Moreover, the novelty of the topic was one of the main reasons the peer review process took longer than usual. Please address all reviewers' suggestions individually and submit your response within 10 (ten) days. Alternatively, you may propose a different timeframe for revisions, provided it is reasonable.

Reviewers' comments:

Reviewer's Responses to Questions

**Comments to the Author**

1. Is the manuscript technically sound, and do the data support the conclusions?

Reviewer #1: Yes

Reviewer #2: Yes

2. Has the statistical analysis been performed appropriately and rigorously?

Reviewer #1: Yes

Reviewer #2: Yes

3. Have the authors made all data underlying the findings in their manuscript fully available?

Reviewer #1: Yes

Reviewer #2: Yes

4. Is the manuscript presented in an intelligible fashion and written in standard English?

Reviewer #1: Yes

Reviewer #2: Yes

Reviewer #1: 1. Line 109-110: The authors mentioned exclusion criteria -- As this is a study about the reliability, I don't see why the authors had to exclude these women. If the study is a part of other studies that require these exclusion criteria, the authors should clarify.

2. Line 118: "(operator 1 and 2, operator 1 and 3, and operator 2 and 3)" gives meaning of each participant receive U/S from 3 sonographers, the authors might consider change to "(operator 1 and 2, operator 1 and 3, 'or' operator 2 and 3)."

3. The authors might consider mention about probe pressure during the measurement.

4. The authors might consider clarify more about the position of the U/S probe for the "cervix-placenta view"

5. Table 2 - misspelling "Wight circumference"

6. The method section mentioned Bland-Altman plots; however, I could not find the plot in the result section. The authors might rephrase the method section to make it align.

Reviewer #2: Thank you for asking me to review the manuscript entitled “Assessing the Reliability of Abdominal Adiposity Measurements by Transabdominal Ultrasound in First Trimester Pregnancies with Sonographers of Varying Experience Levels.”

Abstract: The abstract is well written

Introduction: Well written, however, I wonder why the authors refer to the in-text citation before citing examples (ref 1-2), (ref 3), and (ref 4-6). It is already known that the references are captured in the brackets, so there is no need to add ref to it!

Materials and Methods:

This was well written, however, what advised the choice of statistical analysis used by the authors? Was there a test for normality on the data before deciding which statistical analysis to use?

Did the authors run a positive and negative predictive values test for the study to determine the reliability of Ultrasound in adiposity biometry?

Results:

The result presentation is not very clear!

It started with Table 2? The information in Table 1 can be captured as figures and table 1 should tell us about the demographic characteristics of the study population.

What advised the selection of only 90 participants?

The tables are not self-explanatory, the lines within the tables should be removed!

Discussion:

This is well presented, however, the authors in making reference to intext citations mention both the initial and surnames of the authors like “NJ Kennedy et al. and Nassr AA et al. Which I think is incorrect, it should either be Kennedy et al., or Nassr et al.,.

The discussion is somewhat narrow or shallow, can authors expand on the discussion please?

**Do you want your identity to be public for this peer review?** For information about this choice, including consent withdrawal, please see our Privacy Policy

Reviewer #1: No

Reviewer #2: No

---

## [Author Response · Author response to Decision Letter 1]

27 Mar 2025

Reviewer #1

Comment 1: Line 109-110: The authors mentioned exclusion criteria -- As this is a study about the reliability, I don't see why the authors had to exclude these women. If the study is a part of other studies that require these exclusion criteria, the authors should clarify.

Reply 1: Thank you for your comment. This study is part of a larger prospective study evaluating the predictive ability of first-trimester ultrasound measurements of maternal adiposity for predicting gestational diabetes mellitus. Therefore, specific exclusion criteria were applied to minimize potential confounding effects on outcomes.

Change in the text: No changes were made, but we have clarified this point in our response.

Comment 2: Line 118: "(operator 1 and 2, operator 1 and 3, and operator 2 and 3)" gives meaning of each participant receive U/S from 3 sonographers, the authors might consider change to "(operator 1 and 2, operator 1 and 3, 'or' operator 2 and 3)."

Reply 2: We have revised the wording for improved clarity.

Change in the text: Updated in the Materials and Methods section (lines 118).

Comment 3: The authors might consider mention about probe pressure during the measurement.

Reply 3: We appreciate this suggestion. We have added a statement indicating that ultrasound measurements were taken with care to avoid excessive probe pressure, which could artificially compress abdominal tissue.

Change in the text: Added this detail in the Materials and Methods section (lines 123-124).

Comment 4: The authors might consider clarify more about the position of the U/S probe for the "cervix-placenta view"

Reply 4: Thank you for your comment. We have included additional details regarding the ultrasound probe positioning for the "cervix-placenta view."

Change in the text: Added in Table 1.

Comment 5: Table 2 - misspelling "Wight circumference"

Reply 5: We appreciate your careful review. The spelling error has been corrected.

Change in the text: Updated in Table 2.

Comment 6: The method section mentioned Bland-Altman plots; however, I could not find the plot in the result section. The authors might rephrase the method section to make it align.

Reply 6: Thank you for pointing this out. The Bland-Altman plots were used initially to assess reliability, followed by modifications to establish limits of agreement, which represent the variability range in differences and indicate test-retest reliability for 95% of the population. We have revised the text for clarity.

Change in the text: Updated in Materials and Methods section (lines 133-136).

Reviewer # 2

Comment for introduction part:

The authors refer to in-text citations using "ref" before the bracketed numbers (e.g., "ref 1-2"), which is unnecessary.

Reply: We agree and have corrected all in-text citations to follow the appropriate format using Arabic numerals in square brackets.

Change in the text: Updated throughout the manuscript.

Comment for materials and methods part:

What advised the choice of statistical analysis used by the authors?

Reply: The statistical analysis used in this study was advised and assisted by Dr. Alan Geater, Ph.D., a clinical epidemiology consultant from the Epidemiology Unit, Faculty of Medicine, Prince of Songkla University. This contribution is acknowledged in the Acknowledgements section.

Change in the text: No changes needed, as this is mentioned in Acknowledgements.

Was the data tested for normality before deciding on the statistical analysis?

Reply: Yes, normality testing was performed before statistical analysis to ensure the appropriateness of the methods used.

Change in the text: No changes needed, but we confirm this in our response.

Did the authors run a positive and negative predictive values test for the study to determine the reliability of Ultrasound in adiposity biometry?

Reply: Our objective was to evaluate the reliability of sonographic measurements of abdominal adiposity during the first trimester of pregnancy, not to assess their predictive value for other outcomes. Therefore, we did not perform positive or negative predictive value tests in this study.

Change in the text: No changes needed, but we confirm this in our response.

Comment for results part:

The numbering of tables starts at Table 2. The authors should ensure Table 1 presents demographic characteristics, while other information could be better presented as figures.

Reply: Thank you for your suggestion. In our opinion, Table 1, which describes the standard protocol for measuring each parameter, provides necessary details and is best suited for a tabular format.

Change in the text: No changes were made.

What advised the selection of only 90 participants?

Reply: Our objective was to evaluate intraobserver variability among operators with different experience levels. Each participant was independently scanned by two operators in the following pairings: (operator 1 and 2, operator 1 and 3, or operator 2 and 3). A sample size of 30 participants per operator pair was deemed sufficient to test reliability.

Change in the text: No changes were made, but we clarify this point in our response.

The tables are not self-explanatory, the lines within the tables should be removed!

Reply: Thank you for your suggestion. We have reformatted the tables accordingly.

Change in the text: Updated table formatting.

Comment for discussion part:

The authors refer to in-text citations using both initials and surnames (e.g., "NJ Kennedy et al." and "Nassr AA et al."). This should be corrected to "Kennedy et al." and "Nassr et al."

Reply: We appreciate your attention to detail. The in-text citations have been corrected accordingly.

Change in the text: Updated in the Discussion section (lines 206-207).

The discussion is somewhat narrow or shallow, can authors expand on the discussion please?

Reply: Thank you for your suggestion. As the main objective of this study was to evaluate the reliability of sonographic measurements of abdominal adiposity during the first trimester of pregnancy, we found limited previous studies in this field. While our discussion may appear concise, we believe this study provides a novel foundation for further global research in this area.

Change in the text: No changes were made.

---

## [Decision Letter · Decision Letter 1]

Dear Dr. pruksanusak,

Thank you for submitting your manuscript to PLOS ONE. After careful consideration, we feel that it has merit but does not fully meet PLOS ONE’s publication criteria as it currently stands. Therefore, we invite you to submit a revised version of the manuscript that addresses the points raised during the review process.

One reviewer has recommended rejecting your manuscript due to methodological concerns and questions regarding the generalizability of your findings to clinical practice, while another has advised major revision. In this context, you are encouraged to address these concerns thoroughly before your manuscript can proceed to the next stage of evaluation.

We look forward to receiving your revised manuscript.

Kind regards,

Athanasios G. Pantelis

Academic Editor

PLOS ONE

Reviewers' comments:

Reviewer's Responses to Questions

**Comments to the Author**

Reviewer #3: (No Response)

Reviewer #4: (No Response)

Reviewer #5: All comments have been addressed

Reviewer #6: (No Response)

2. Is the manuscript technically sound, and do the data support the conclusions?

Reviewer #3: Yes

Reviewer #4: Yes

Reviewer #5: Yes

Reviewer #6: No

3. Has the statistical analysis been performed appropriately and rigorously?

Reviewer #3: (No Response)

Reviewer #4: Yes

Reviewer #5: Yes

Reviewer #6: Yes

4. Have the authors made all data underlying the findings in their manuscript fully available?

Reviewer #3: Yes

Reviewer #4: Yes

Reviewer #5: Yes

Reviewer #6: Yes

5. Is the manuscript presented in an intelligible fashion and written in standard English?

Reviewer #3: Yes

Reviewer #4: Yes

Reviewer #5: Yes

Reviewer #6: Yes

Reviewer #3: I have 2 main concern: why did you select the first trimester? was it based on previous reports? we know that the fat deposits increase through second and third trimester, and these are the periods where pregnant women develop the actual insulin resistance. The second concern is the age. If you included younger women at 20th for example, this can remove the age factor of fat deposition. Also I wonder why did you select different levels of trainees to do the abdominal scan. As a first report in the country, you should put a solid background for future researches by conducting your study with expert hands. Also why did you excluded hypertensive women, hypertension alone has no impact on fat deposition unless itis related to obesity or diabetes

Reviewer #4: Gestational diabetes mellitus is a significant problem, and one of its risk factors is central obesity. A common tool for assessing body composition (BMI) does not capture central obesity, and thus is an imprecise indicator of the risk of GDM. In this study, the authors use ultrasound to measure central obesity in pregnant Asian people. Their aim in this study is to quantify the extent to which a clinician’s level of experience affects the reliability of ultrasonic measurements of body fat taken in three different areas where fat is deposited in the trunk. The results indicate that the ultrasonic measurements of body fat correlate well with hip-to-waist ratio, hip circumference, and waist circumference. Moreover the degree of intra- and inter-clinician variability in taking these measurements is quite minimal.

The paper is clearly written. The authors clearly and convincingly describe the shortcomings of BMI as a diagnostic tool, and present a convincing case in favor of new techniques to measure fat deposition in the trunk and thus more readily identify women at risk of GDM. The statistical methods appear sound, and the methods are clear. Table 1 is especially strong, as it shows exactly what differentiates the measurements taken at the different abdominal sites. In Tables 3 and 4 it is tricky to discern the most important comparisons the authors wish to draw our attention to. I wonder if every pairwise comparison needs to be presented. A summary could serve well, alternatively, the data in Tables 3 and 4 could perhaps be offered as supplemental materials.

I have one primary concern, which is that the findings of the manuscript on inter-practitioner variability don’t clearly stand on their own. That is, the results described in this manuscript on inter-sonographer variability would be a crucial component of a larger study establishing the validity, relevance, or predictive value of the method itself. Such a methods paper might look at whether or not abdominal ultrasound compared favorably to other methods of measuring body composition such as body densitometry, CT, or MRI. Or, as seems to be the case here (based on my reading of an earlier response to reviewers), such a methods paper might look at the predictive value of ultrasound in assessing the risk of gestational diabetes. But these findings on measurement variability alone are presently separated from a context in which they would become compelling, and reach a wider audience.

Reviewer #5: Summary

This study is significant and has relevance in maintaining the health of both the mother and fetus. The aim and objective of the study were achieved through the different methods, results and the discussion. The study targeted the right subjects and the tools were non-invasive which made the methods more attractive. And elicited low or no discomfort to the pregnant women during their applications.

Abstract

Can the reason why these tools were considered be incorporated in the abstract?

Yes, it is to measure the abdominal adiposity in pregnancy. But why? Line 28…. Let the correlation between the measurement of the abdominal adiposity and gestational diabetes mellitus be incorporated.

Introduction

Line 66…. Gave the reasons why this study was conducted, however, I believe this tool would be used in conjunction with other known methods in determining/diagnosing gestational diabetes mellitus.

Materials and Methods

Line 119 – 122…. What informed you of these 3 locations for the measurements?

Results

Line 153 – 155…. The probability that the procedure and results would be affected by experience of the operators was expected to be high as rightly indicated in this study. However, line 158, interestedly showed that VFT was not affected by experience.

Discussion

Line 181… this sentence should be reworded. ‘For Thai population, one study found similar results to ours with high intraobserver reliability (ICC 0.96)’ …. It could have be written as ‘our result was similar to a previous study which had a high intraobserver reliability’ (ICC 0.96). My understanding is that you compare new study with previous studies.

Figures and Tables: They were simple, self-explanatory, easy to the eye and well labelled.

Reviewer #6: 83- difference ---- it should be different

90-- research and contribute to the development of more effective GDM screening strategies---you are writing in introduction theseu/s technique for screeing of GDM --but already simple and reliable methods are there and only OBESITY is not the risk factor for GDM

118---------operators are not equally distributed as far as experence is concerned

121----- the cervix-placenta view need some explanaition how it measures and detect VF

190---------again your focus on prediction of GDM but can you explain that when other simple methods are availble then with this technique and it is obviuos experenced operator has more relable result so why the article only check the reliabilty of operators

then authour did not mention that how much it has predictive value in screening GDM as your introduction is emaphasiszing on it

**Do you want your identity to be public for this peer review?** For information about this choice, including consent withdrawal, please see our Privacy Policy

Reviewer #3: No

Reviewer #4: No

Reviewer #5: No

Reviewer #6: **Yes: ** professor farzana rizwan arain

---

## [Author Response · Author response to Decision Letter 2]

12 May 2025

Reviewer #3

Comment 1: Why did you select the first trimester? Was it based on previous reports? We know that fat deposits increase through the second and third trimester, and these are the periods where pregnant women develop the actual insulin resistance.

Response: We selected the first trimester because it represents a baseline period before the onset of significant insulin resistance in pregnancy. Early identification of central adiposity may offer predictive insights before metabolic changes occur. Additionally, conducting assessments in the first trimester helps reduce potential confounding from physiological changes occurring later in pregnancy.

Change in the text: We have added justification in the Introduction section (Lines 107-111).

Comment 2: The second concern is the age. If you included younger women at 20th for example, this can remove the age factor of fat deposition.

Response: Thank you for this point. Our inclusion criteria accepted a wide age range to reflect real-world pregnant populations.

Change in the text: No changes were made, but we have clarified this point in our response.

Comment 3: Why did you select different levels of trainees to do the abdominal scan? As a first report in the country, you should put a solid background for future research by conducting your study with expert hands.

Response: Our study aimed to evaluate the reliability of sonographic measurements across varying operator experience levels, as this reflects actual clinical scenarios, particularly in resource-limited settings. We now clarify this rationale in the Methods and Discussion sections.

Change in the text: No changes were made, but we have clarified this point in our response.

Comment 4: Why did you exclude hypertensive women? Hypertension alone has no impact on fat deposition unless related to obesity or diabetes.

Response: We excluded women with hypertension to minimize confounding effects, as chronic hypertension may independently influence vascular and metabolic status. Moreover, ACOG Practice Bulletin No. 190 identifies hypertension as a risk factor for gestational diabetes mellitus, supporting our decision to exclude such cases.

Change in the text: No changes were made, but we have clarified this point in our response.

Reviewer #4

Comment 1: Tables 3 and 4 are hard to interpret. Could a summary be used or move data to supplementary materials?

Response: Thank you for the suggestion. Tables 3 and 4 present key findings regarding intraobserver and interobserver variability, which directly relate to the primary objective of our study. Therefore, we believe they should remain in the main manuscript. However, we have added a concise summary of the main findings in the Results section (Lines 174-180).

Change in the text: No changes were made, but we have clarified this point in our response.

Comment 2: Findings on inter-sonographer variability do not stand on their own; suggest linking to broader study of validity or predictive value.

Response: We appreciate this valuable feedback. We have now clarified in the Introduction and Discussion that this study serves as a foundational methodological assessment of measurement reliability. Future studies will build on this by evaluating predictive value using standard diagnostic outcomes.

Change in the text: No changes were made, but we have clarified this point in our response.

Reviewer #5

Comment 1 (Abstract): Why were these tools considered? Line 28: Let the correlation between abdominal adiposity and GDM be incorporated.

Response: We revised the abstract to clearly indicate that the study was motivated by the potential association between abdominal adiposity and GDM risk.

Change in the text: Updated in the Abstract section (Lines 27-29).

Comment 2 (Introduction): Line 66: This tool would be used alongside other methods for diagnosing GDM.

Response: The section was revised to reflect that sonographic assessment of adiposity is a complementary tool, not a replacement for current GDM screening strategies.

Change in the text: Updated in the Introduction section (Lines 91-93 and 116-120).

Comment 3 (Methods): Lines 119–122: What informed you of the 3 locations for measurement?

Response: The three measurement sites were selected based on prior studies and anatomical landmarks representing distinct fat compartments.

Change in the text: We have added explanation and references in Materials and Methods (Lines 140) and Table 1.

Comment 4 (Results): Line 153 – 155: The probability that the procedure and results would be affected by experience of the operators was expected to be high as rightly indicated in this study. However, line 158, interestedly showed that VFT was not affected by experience.

Response: This observation is now more clearly emphasized in the Results section. We highlight that the ICC indicated excellent reliability across all adiposity parameters, with only the VFT measurement between the most and least experienced operators showing slightly lower, but still good, reliability (Lines 174-180).

Change in the text: No changes were made, but we have clarified this point in our response.

Comment 5 (Discussion): Line 181: Suggest rewording to compare clearly with previous studies.

Response: We revised the sentence to: “Our result was similar to a previous study which had a high intraobserver reliability (ICC 0.96).”

Change in the text: Updated in the Discussion section (Lines 202-203).

Reviewer #6

Comment 1: Line 83: “difference” should be “different”

Response: This has been corrected.

Change in the text: Updated in the Introduction section (Lines 109).

Comment 2: Line 90: Introduction suggests ultrasound is for GDM screening, but simple methods exist and obesity is not the only risk factor.

Response: We revised the Introduction to clarify that our aim is not to replace existing screening methods, but to explore the potential of ultrasound as a supportive tool for early risk identification based on adiposity.

Change in the text: Updated in the Introduction section (Lines 116-120).

Comment 3: Line 118: Operators are not equally distributed by experience.

Response: We acknowledge this and explain that our study design reflects real-world clinical practice, particularly in resource-limited settings, where varying levels of operator experience are common. Our study thus aimed to evaluate measurement reliability in such settings.

Change in the text: No changes were made, but we have clarified this point in our response.

Comment 4: Line 121: “Cervix-placenta view” needs explanation for how it measures VFT.

Response: Thank you for the comment. The cervix-placenta view (technique 2), as described in Table 1, was used only to measure SFT. VFT was measured using technique 1, at the level of the linea alba, 1–2.5 cm above the umbilicus (Lines 140-144).

Change in the text: No changes were made, but we have clarified this point in our response.

Comment 5: Line 190: Why focus only on operator reliability when prediction of GDM is emphasized in the Introduction?

Response: While the introduction highlights the potential association between abdominal adiposity and the risk of GDM, the primary objective of this study was to assess the reliability of sonographic measurements of abdominal adiposity in the first trimester. Assessing predictive value for GDM would require longitudinal follow-up, which are beyond the scope of this methodological assessment. We now clarify this point in the text.

Change in the text: No changes were made, but we have clarified this point in our response.

---

## [Editor Report · Decision Letter 2]

Assessing the Reliability of Abdominal Adiposity Measurements by Transabdominal Ultrasound in First Trimester Pregnancies with Sonographers of Varying Experience Levels

PONE-D-24-49281R2

Dear Dr. Pruksanusak,

We’re pleased to inform you that your manuscript has been judged scientifically suitable for publication and will be formally accepted for publication once it meets all outstanding technical requirements.

Kind regards,

Athanasios G. Pantelis

Academic Editor

PLOS ONE

Additional Editor Comments (optional):

Dear Dr. Pruksanusak and colleagues,

Thank you for your detailed responses to the reviewers’ comments and for submitting the revised version of your manuscript titled: “Assessing the reliability of abdominal adiposity measurements by transabdominal ultrasound in first trimester pregnancies with sonographers of varying experience levels” (PONE-D-24-49281R2).

I appreciate the effort you put into addressing the reviewers’ concerns. Your clarifications and the added content significantly enhance the manuscript’s clarity and value, particularly in reinforcing its methodological contribution to the field.

I would like to offer my sincere apologies for the prolonged editing time and thank you for your patience throughout the process.

Before proceeding to final acceptance, I kindly ask that you make a few minor language corrections to address small typographical and grammatical issues noted in the current version (e.g., “aimsT” in the Abstract, redundant phrasing in the Introduction, missing punctuation in the Discussion).

Once these final edits are incorporated, your manuscript will be ready for publication.

---

## [Editor Report · Acceptance letter]

PONE-D-24-49281R2

PLOS ONE

Dear Dr. pruksanusak,

I'm pleased to inform you that your manuscript has been deemed suitable for publication in PLOS ONE. Congratulations! Your manuscript is now being handed over to our production team.

Kind regards,

on behalf of

Dr. Athanasios G. Pantelis

Academic Editor

PLOS ONE